# MEASURING FAIRNESS USING PROBABLE SEGMENTATION FOR CONTINUOUS SENSITIVE ATTRIBUTES

## ABSTRACT

Algorithmic fairness in machine learning aims to regulate the bias towards sensitive attributes. In the case of continuous sensitive attributes, however, defining and measuring fairness is a non-trivial task. For instance, estimating a maximum disparity of predictions within a continuous sensitive attribute is vulnerable for an extreme case, whereas a mean disparity of predictions underestimates the effect of the worst case, which is meaningful for testing the independence between the prediction and the sensitive attribute. We address this issue by developing a new definition of fairness, probable demographic parity, based on a maximum prediction disparity of probable segments. We only consider probable segments of the continuous sensitive attribute that have a higher probability than the preset minimum probability condition. Then, we compare the local prediction average of these segments to identify the maximum prediction disparity. By doing so, we ensure a consistent estimation error for the local prediction average of the segment and mitigate the risk of encountering missing data in the segment. We analyze the various theoretical features including stability and independence and experimentally prove the usefulness of the proposed metric.

## 1 INTRODUCTION

A machine learning model is vulnerable to bias where the dataset is biased with respect to sensitive attributes such as gender or age (Calders et al., 2009; Kamiran & Calders, 2012). Fairness has drawn substantial attention along with this situation in machine learning since fairness aims to seek the optimal model while alleviating the unjustifiable prediction of the machine learning model (Kamishima et al., 2012; Pleiss et al., 2017; Donini et al., 2018; Agarwal et al., 2018). To achieve the goal of fairness, it is an important challenge to accurately quantify fairness. For instance, for demographic parity which is one the definitions of fairness ensuring the independence of the prediction and the sensitive attribute (Dwork et al., 2012; Zafar et al., 2017; Madras et al., 2018; Mehrabi et al., 2021), the fairness metrics based on a maximum disparity of predictions within the sensitive attribute (Alabdulmohsin & Lucic, 2021; Denis et al., 2021; Alabdulmohsin et al., 2022) or the fairness metric based on a mean disparity of the predictions within the sensitive attribute (Cho et al., 2020) has been studied.

Amounts of fairness metrics assume that there are discrete sensitive attributes, which are only available under limited environments (Mary et al., 2019; Grari et al., 2019). On the other hand, the fairness metric for continuous sensitive attributes such as Mary et al. (2019); Creager et al. (2019) has been proposed based on a statistical test of independence of the prediction and the sensitive attribute. Furthermore, Jiang et al. (2021) propose a computationally tractable metric based on a mean disparity of predictions within continuous sensitive attributes. However, the mean prediction disparity-based fairness metric suffers from the underestimation problem and causes the gerrymandering problem (Kearns et al., 2018). As shown in Figure 1 (a), although the left and the right extreme segments' predictions show the violation of independence between the prediction $\hat{Y}$ and the sensitive attribute $S$, the effects of these segments are underestimated at the mean disparity-based fairness metric due to the low probability of the extreme segments. One possible solution for this problem is to use the fairness metric based on a maximum prediction disparity as Alabdulmohsin et al. (2022). Contrary to the mean disparity-based fairness metrics, however, the maximum disparity-based fairness metrics are vulnerable to the outlier and easily suffer from a large estimation variance concerning the low probability segments illustrated in Figure 1 (b).

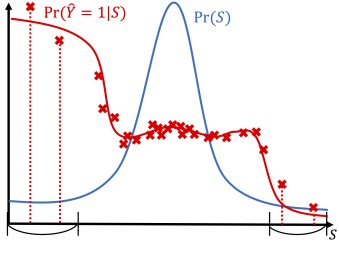 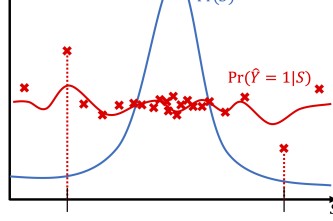

(a) Underestimated segments for
mean disparity based fairness metrics

(b) Overestimated segments for
maximum disparity based fairness metrics

Figure 1: Illustrations for an underestimation and overestimation problem at the low probability segments. The lines denote the true distributions, and the x marks denote estimations.

We address this under- and over-estimation trade-off of continuous sensitive attributes by introducing *probable demographic parity*. Probable demographic parity is defined as the invariance across the local prediction averages of the probable segments that have at least the predefined probability $\alpha$ within the continuous sensitive attribute. Equivalently, probable demographic parity implies the independence between predictions and the probable segment of the sensitive attribute. We present the fairness metric based on a maximum violation of probable demographic parity and also present the empirical estimation method. For the empirical estimation, we approximate the probable segments as the segments that include at least $N\alpha$ samples since the estimated probability of the segments should be greater than or equal to $\alpha$ where $N$ is the total number of the samples. By doing so, the proposed fairness metric handles the overestimation problem by leveraging the stability of the mean operation within the probable segment. That is, although we encounter the data sparsity region within the sensitive attribute, we adaptively segment the sensitive attribute so that we can estimate the prediction of the segment with a reliable error. Furthermore, the proposed fairness metric alleviates the underestimation problem since the metric takes into account the maximum violation of parity. Probable demographic parity also preserves tractable computation, unlike the fairness metrics based on a statistical metric such as Mary et al. (2019); Creager et al. (2019).

Probable demographic parity can be interpreted as an extension of demographic parity. For discrete sensitive attributes, probable demographic parity is equivalent to demographic parity when every probability of the discrete sensitive attribute's outcome is greater than or equal to the given $\alpha$. In addition, for continuous sensitive attributes, we present the connection between probable demographic parity and demographic parity under the Lipschitz condition or quantized continuous sensitive attribute. Likewise, probable demographic parity is closely related to demographic parity.

In summary, we propose probable demographic parity to measure fairness for continuous sensitive attributes. Building upon the novel definition of probable segments, probable demographic parity harmonizes the strengths of the preceding methods including the mean and maximum prediction disparity-based fairness metrics. In addition, we reveal the connection between probable demographic parity and demographic parity. We present the empirical estimation algorithm for probable demographic parity and analyze an empirical error and computational complexity to demonstrate its usefulness. Although we mainly focus on demographic parity for a binary classification task, we extend our metric for a regression task, and the proposed metric can be easily extended to other fairness definitions such as equalized odds.

## 2 BACKGROUND

We consider a binary classification task with the input $X \in \mathbb{R}^d$, the label $Y \in \{0, 1\}$, the prediction $\hat{Y} \in \{0, 1\}$, and the sensitive attribute $S$. Given the discrete sensitive attribute $S \in \{0, .., K\}$, demographic parity (DP) is satisfied when the following holds:

$$\forall s, s' \in \{0, .., K\}, \ \Pr(\hat{Y} = 1 | S = s) = \Pr(\hat{Y} = 1 | S = s').$$

We measure fairness for a discrete sensitive attribute using the violation of demographic parity such as Dwork et al. (2012); Zafar et al. (2017); Madras et al. (2018); Mehrabi et al. (2021). Formally,

the fairness is measured as follows:

$$\Delta_{DP} := \max_s \Pr(\hat{Y} = 1 | S = s) - \min_s \Pr(\hat{Y} = 1 | S = s). \tag{1}$$

To overcome the difficulty of optimization, we can leverage the relaxed fairness metric as follows:

$$\Delta_{DP} := \max_s \mathbb{E}_X[f(X)|S = s] - \min_s \mathbb{E}_X[f(X)|S = s]. \tag{2}$$

We can directly extend the fairness metric $\Delta_{DP}$ for the continuous sensitive attribute $S \in [0, 1]$. However, the metric based on the maximum disparity is vulnerable. They overestimate the significance of extreme cases although the argument of maximum or minimum at the sensitive attribute is not probable in real situations since the argument has zero probability. Alternatively, we can leverage the fairness metric based on a mean prediction disparity within the continuous sensitive attribute. For instance, Jiang et al. (2021) proposes generalized demographic parity (GDP) using the mean disparity between local and global mean weighted by probability:

$$\Delta_{GDP} := \mathbb{E}_S[|\mathbb{E}_X[f(X)|S] - \mathbb{E}_X[f(X)]|]. \tag{3}$$

Unfortunately, the mean disparity-based fairness metric may underestimate the effect of the low probability region of the sensitive attribute and even characteristics of the arguments of maximum or minimum at the sensitive attribute can be regarded as insignificant. In this work, to address the aforementioned challenges, we propose a composition of both approaches called probable demographic parity.

**Related works**   Fairness has been measured by the two major definitions, group fairness and individual fairness. Group fairness focuses on the invariance of the predictions to a protected attribute including demographic parity (Feldman et al., 2015), equalized odds (Hardt et al., 2016), and equal opportunity (Hardt et al., 2016), while individual fairness focuses on the similarity of the predictions for similar individuals (Dwork et al., 2012; Garg et al., 2019; Friedler et al., 2021). In this work, we focus on the group fairness metrics for continuous sensitive attributes such as Mary et al. (2019); Creager et al. (2019) to mitigate computational intractability and instability problems.

## 3   METHOD

In this section, we introduce probable demographic parity to measure fairness and present an empirical method to estimate probable demographic parity.

### 3.1   PROBABLE DEMOGRAPHIC PARITY

For probable demographic parity, we consider probable segments of the sensitive attribute which has a higher probability than the hyperparameter $\alpha \in (0, 1)$. By doing so, we mitigate the unreliability of the maximum prediction disparity-based fairness metric. Given the continuous sensitive attribute $S \in [0, 1)$, we define the probable segment $[k, k')$ of attributes $S$ as a segment which has a higher probability than the hyperparameter $\alpha$, i.e., $[k, k')$ is a probable segment when $\int_k^{k'} \Pr(S = s)\mathrm{d}s \geq \alpha$ or $(F(k') - F(k)) \geq \alpha$ where $F(\cdot)$ denotes a cumulative distribution function defined as $F(s) = \Pr(S \leq s)$. To alleviate the computational cost and eliminate redundant segments, we define a minimally probable segment $[k, h_\alpha(k))$ using a minimally probable segmentation function $h_\alpha(\cdot)$ as follows:

$$h_\alpha(k) := \arg\min_{k'} \int_k^{k'} \Pr(S = s)\mathrm{d}s \geq \alpha, \tag{4}$$

where $[k, k') \subset [0, 1]$. Also, for simplicity, a local prediction average of a segment $[k, k') \subset [0, 1]$ is defined as follows:

$$M_{X,S}([k, k')) := \mathbb{E}_X[f(X)|k \leq S < k']$$

$$= \frac{\int_k^{k'} \mathbb{E}_X[f(X)|S = s] \Pr(S = s)ds}{\int_k^{k'} \Pr(S = s)ds}. \tag{5}$$

Finally, we define probable demographic parity as parity across the minimally probable segments. Formally,

$$\forall [k, h_\alpha(k)), [k', h_\alpha(k')) \subset [0, 1],$$
$$M_{X,S}([k, h_\alpha(k))) = M_{X,S}([k', h_\alpha(k'))). \tag{6}$$

It is worth noting that probable demographic parity denotes the independence between the prediction and the prominent segment which consists of minimally probable segments with the minimum probability $\alpha$. Then, we measure fairness using the violation of probable demographic parity based on the maximum disparity of minimally probable segments with probability $\alpha$ as follows:

$$\Delta_{\alpha\text{-}DP} := \max_{[k, h_\alpha(k))} M_{X,S}([k, h_\alpha(k))) - \min_{[k', h_\alpha(k'))} M_{X,S}([k', h_\alpha(k'))),$$
$$\text{where } [k, h_\alpha(k)), [k', h_\alpha(k')) \subset [0, 1]. \tag{7}$$

By using the proposed fairness metric based on probable demographic parity, we derive the maximum disparity of prominent segments while preserving the advantage of the mean disparity-based metric.

### 3.1.1 PROBABLE DEMOGRAPHIC PARITY FOR DISCRETE SENSITIVE ATTRIBUTES

The proposed fairness metric, probable demographic parity, is also available under discrete sensitive attributes as well as continuous sensitive attributes since probable demographic parity is an extension of demographic parity. Given the discrete sensitive attribute $S \in \{1, ..., K\}$, we define a minimally probable segmentation function $h_\alpha^d(k)$ as follows:

$$h_\alpha^d(k) := \arg\min_{k'} \sum_{s=k}^{k'} \Pr(S = s) \geq \alpha. \tag{8}$$

Also, a local prediction average of the segment $\{k, ..., h_\alpha^d(k)\}$ of the discrete sensitive attribute $S$ is defined as follows:

$$M_{X,S}^d(\{k, ..., k'\}) := \mathbb{E}_X[f(X)|k \leq S \leq k']$$
$$= \frac{\sum_{s=k}^{k'} \mathbb{E}_X[f(X)|S = s] \Pr(S = s)}{\sum_{s=k}^{k'} \Pr(S = s)}. \tag{9}$$

Finally, we measure fairness for a discrete sensitive attribute using probable demographic parity as follows:

$$\Delta_{\alpha\text{-}DP}^d := \max M_{X,S}^d(\{k, ..., h_\alpha^d(k)\}) - \min M_{X,S}^d(\{k', ..., h_\alpha^d(k')\}),$$
$$\text{where } \{k, ..., h_\alpha^d(k)\}, \{k', ..., h_\alpha^d(k')\} \subset \{0, ..., K\}. \tag{10}$$

**Theorem 1.** *Given $\alpha$, for the discrete sensitive attribute $S \in \{1, ..., K\}$, if $S$ satisfies $\forall s \in \{1, ..., K\}$, $\Pr(S = s) \geq \alpha$, then $\Delta_{\alpha\text{-}DP}^d = \Delta_{DP}$.*

Theorem 1 can be easily shown using the definition of a minimally probable segment because each minimally probable segment only includes a single component, i.e., $\forall s \in \{1, ..., K\}$, $h_\alpha^d(s) = s$. However, note that Theorem 1 does not imply if and only if statement.

### 3.2 ESTIMATION OF PROBABLE DEMOGRAPHIC PARITY

We propose an empirical estimation method for our metric. Since the true $\Pr(S)$ is unknown, we estimate the minimally probable segmentation function $h_\alpha(k)$ for the first stage. Given the tuples of samples $(x_n, s_n, y_n)_{n=1}^N$ consisting of the input $x_n$, the label $y_n$, and the attribute $s_n$, $h_\alpha(k)$ can be estimated by approximating the density as $\int_k^{k'} \Pr(S = s)\mathrm{d}s = F(k') - F(k) \approx \frac{1}{N} \sum_{n=1}^N \mathbf{I}_{k \leq s_n < k'}$, where $\mathbf{I}_{k \leq s_n < k'}$ is an indicator function. Finally, we estimate probable demographic parity using an estimated minimally probable segmentation function $\tilde{h}_\alpha(k)$ and local prediction average $\tilde{M}_{X,S}([k, k'))$ which are defined as follows:

$$\tilde{h}_\alpha(k) := \arg\min_{k'} \frac{1}{N} \sum_{n=1}^N \mathbf{I}_{k \leq s_n < k'} \geq \alpha,$$
$$\tilde{M}_{X,S}([k, k')) := \frac{\sum_n \mathbf{I}_{k \leq s_n < k'} f(x_n)}{\sum_n \mathbf{I}_{k \leq s_n < k'}}, \tag{11}$$
$$\tilde{\Delta}_{\alpha\text{-}DP} := \max_k \tilde{M}_{X,S}([k, \tilde{h}_\alpha(k))) - \min_{k'} \tilde{M}_{X,S}([k', \tilde{h}_\alpha(k'))).$$

---

**Algorithm 1** Empirical Probable Demographic Parity

---

**Input:** Dataset $(x_n, s_n, y_n)_{n=1}^N$, minimum probability $\alpha$
1: **begin**
2:   // Preparation //
3:   $(x_r, s_r, y_r)_{r=1}^N = Sort((x_n, s_n, y_n)_{n=1}^N, \text{key} = s_n)$
4:   Cache whether $s_r = s_{r+1}$ or $s_r \neq s_{r+1}, \forall r \in \{1, ..., N-1\}$
5:   $r^{(0)} \leftarrow 0, idx^{(-1)} \leftarrow \emptyset, S^{(-1)} \leftarrow 0$
6:   **for** $(i \leftarrow 0; \frac{N - r^{(i)} + 1}{N} \geq \alpha; i \leftarrow i+1)$ **do**
7:     // Estimate a minimally probable segmentation function //
8:     $t^{(i)} \leftarrow \arg\min_r \dfrac{r - r^{(i)} + 1}{N} \geq \alpha$ and $s_r \neq s_{r+1}$
9:     $\tilde{h}_\alpha(s_{r^{(i)}}) \leftarrow s_{t^{(i)}}$
10:     // Estimate a local prediction average //
11:     $idx^{(i)} \leftarrow \{r\}_{r=r^{(i)}}^{t^{(i)}}$
12:     $S^{(i)} \leftarrow S^{(i-1)} - \sum_{r \in idx^{(i-1)} \setminus idx^{(i)}} f(x_r) + \sum_{r \in idx^{(i)} \setminus idx^{(i-1)}} f(x_r)$
13:     $M^{(i)} \leftarrow \dfrac{S^{(i)}}{t^{(i)} - r^{(i)} + 1}$
14:     $r^{(i+1)} \leftarrow \arg\min_{r > r^{(i)}} s_r \neq s_{r^{(i)}}$
15: **end for**
**Output:** $\tilde{\Delta}_{\alpha\text{-}DP} = \max_i M^{(i)} - \min_i M^{(i)}$

---

Since every estimated probable segment includes at least $N\alpha$ samples by the definition, the proposed method does not suffer from the segment that includes zero samples, and the variance is reduced. Details are explained in Section 4.1.

### 3.2.1 COMPUTATIONAL COMPLEXITY

A detailed algorithm for empirical probable demographic parity is introduced in Algorithm 1. The algorithm consists of a preparation stage such as sorting and caching, estimation of a minimally probable segmentation function, and estimation of a local prediction average. Although the segments are different, the local prediction averages of the segments are identical if the different segments include identical samples. Thus, although $k$ is a continuous variable, discrete $k$ such that $k = s_r$ for $r = 1, ..., N$ are considered. Also, by only considering "minimally" probable segments, we reduce computational complexity from $O(N^2)$ to $O(N)$ by fixing $t^{(i)}$ as a single value for each $i$. When estimating a local prediction average of a probable segment, calculating an average for every segment results in a complexity of $O(N^2)$. However, if we cache a previous summation value $S^{(i-1)}$, each sample is accessed only two times including one addition and one subtraction, and the complexity is reduced to $O(N)$. Consequently, given the sorted samples, the total computational complexity is retained as $O(N)$.

## 4 ANALYSIS

### 4.1 ESTIMATION ERROR ANALYSIS

The probable segments are empirically estimated for probable demographic parity, and subsequently, the local prediction averages of the probable segments are estimated. In this section, we investigate the two types of errors including the estimation of probable segments and the estimation of a local prediction average.

### 4.1.1 PROBABLE SEGMENTATION

We estimate a probable segment using the number of samples in a segment. If the segment includes $k$ samples, then the estimated probability of the segment is $k/N$ where $N$ denotes the total number of samples. Herein, we provide the rationale for this design choice based on maximum likelihood estimation. For a segment in the sensitive attribute with a probability $p$, the probability that $k$ samples

are included in the given segment follows binomial distribution $B(N, p)$. By maximum likelihood estimation of $B(N, p)$, $p = k/N$ is the optimal probability for unknown $p$, which supports our probability estimation.

### 4.1.2 LOCAL PREDICTION AVERAGE

After the probable segment estimation stage, we estimate the local prediction average for $\tilde{\Delta}_{\alpha-DP}$.

**Theorem 2.** $\tilde{M}_{X,S}([k, \tilde{h}_\alpha(k)))$, *the local prediction average for the estimated minimally probable segments* $[k, \tilde{h}_\alpha(k))$, *satisfies the following:*

$$Bias(\tilde{M}_{X,S}([k, \tilde{h}_\alpha(k)))) = 0. \tag{12}$$

*Also, where* $\sigma^2_{max}$ *is the maximum variance across an empirical minimally probable segment as* $\sigma^2_{max} = \max_k Var(f(X) \mid k \leq S < \tilde{h}_\alpha(k))$, *the variance of the local prediction average is bounded as:*

$$Var(\tilde{M}_{X,S}([k, \tilde{h}_\alpha(k)))) \leq \frac{\sigma^2_{max}}{N\alpha}. \tag{13}$$

The estimated minimally probable segment $[k, \tilde{h}_\alpha(k))$ includes at least $N\alpha$ samples by the estimation algorithm. Thus, the estimation of local prediction average $\tilde{M}_{X,S}([k, \tilde{h}_\alpha(k)))$ converges to the real average as $N$ grows large, and the variance is divided by $N\alpha$. That is, the local prediction average is stable across the probable segments unlike the equally spaced segmentation in which the segment has identical length but the number of samples is not assured. The experimental result for this argument is discussed in Section 5.1.

### 4.2 COMPARISON OF PROBABLE DEMOGRAPHIC PARITY AND DEMOGRAPHIC PARITY

Demographic parity requires the independence of the prediction $\hat{Y}$ and the sensitive attribute $S$. Therefore, we theoretically examine probable demographic parity and traditional demographic parity to explore the connection between a test of independent and probable demographic parity.

### 4.2.1 QUANTIZED SENSITIVE ATTRIBUTES

**Theorem 3.** *Given* $\alpha$, *for every minimally probable segments set* $R = \{seg^\alpha_{(i)}\}_{i=1}^K$ *where* $\{seg^\alpha_{(i)}\}_{i=1}^K$ *is a partition of the continuous sensitive attribute* $S \in [0, 1)$, *i.e.,* $\bigcup_{i=1}^K seg^\alpha_{(i)} = [0, 1)$ *and* $\forall i, j \in \{1, ..., K\}(j \neq i)$, $seg^\alpha_{(i)} \bigcap seg^\alpha_{(j)} = \emptyset$, *define the quantization function* $Q : [0, 1) \to \{1, ..., K\}$ *such that* $Q(S) = i$ *if* $S \in seg^\alpha_{(i)}$. *If we have* $f$ *with* $\Delta_{\alpha\text{-}DP} = \delta$ *with respect to* $S$, *then* $\Delta_{DP}$ *and* $\Delta^d_{\alpha\text{-}DP}$ *with respect to the discrete sensitive attribute* $Q(S)$ *quantized from the continuous sensitive attribute* $S$ *satisfies* $\Delta_{DP} = \Delta^d_{\alpha\text{-}DP} \leq \delta$.

Theorem 3 shows the connection between probable demographic parity and demographic parity. Moreover, if $f$ satisfies probable demographic parity, i.e., $\Delta_{\alpha\text{-}DP} = 0$, then $f$ satisfies $\Delta_{DP} = 0$ with respect to $Q(S)$, which satisfies the demographic parity condition. That is, Theorem 3 reveals the connection between probable demographic parity and a test of independence by bridging probable demographic parity and demographic parity of the quantized sensitive attribute utilizing minimally probable segments. The proof is provided in Appendix A.1.

### 4.2.2 LIPSCHITZ CONDITION

A prediction $\mathbb{E}_X[f(X)|S = s]$ is said to be $\gamma$-Lipschitz if there exists $\gamma > 0$ such that $|\mathbb{E}_X[f(X)|S = s] - \mathbb{E}_X[f(X)|S = s']| \leq \gamma|s - s'|$.

**Theorem 4.** *Assume that the prediction* $\mathbb{E}_X[f(X)|S]$ *for the continuous sensitive attribute* $S \in [0, 1)$ *satisfies the* $\gamma$-*Lipschitz condition. Then the distance between the fairness metric* $\Delta_{\alpha\text{-}DP}$ *based on probable demographic parity for* $S$ *and the fairness metric* $\Delta_{DP}$ *based on demographic parity for* $S$ *is bounded as:*

$$|\Delta_{DP} - \Delta_{\alpha\text{-}DP}| \leq 2\gamma \max_k |h_\alpha(k) - k|. \tag{14}$$

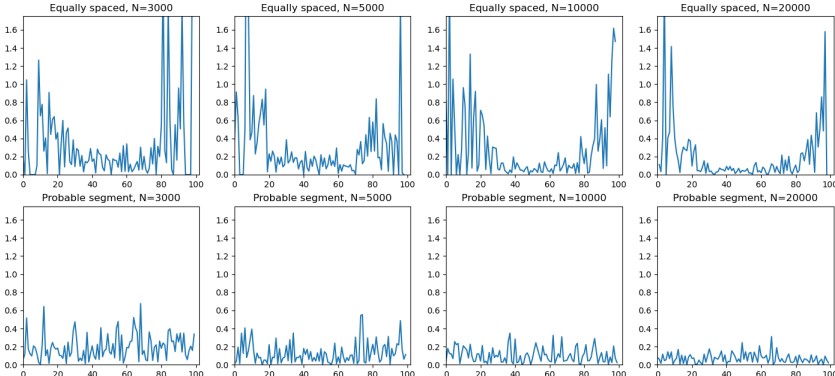

Figure 2: The error between the real and the estimated local prediction average for both segmentation methods. The $x$-axis denotes a segment number and the $y$-axis denotes an estimation error of each segment. The upper figures indicate the equally spaced segmentation method and the lower figures indicate the probable segmentation method.

We compare probable demographic parity to demographic parity under the Lipschitz condition. Theorem 4 confirms that the distance between demographic parity metric $\Delta_{DP}$ and the probable demographic parity metric $\Delta_{\alpha\text{-}DP}$ is bounded under the Lipschitz condition. That is, if we achieve an acceptable performance in terms of probable demographic parity, then demographic parity can also be competent. When $\alpha$ is decreased, $h_\alpha(k)$ is equals to or decreased following $\alpha$, so, the difference of $\Delta_{DP}$ and $\Delta_{\alpha\text{-}DP}$ is diminished. The proof is provided in Appendix A.2.

### 4.3 EXPANDING TO ADDITIONAL FAIRNESS METRICS AND TASKS

In this paper, we focus on a binary classification task for fairness. However, we can directly extend our method to continuous label environments (i.e., a regression task) or a multi-class classification task. As an example, we apply probable demographic parity for a regression task and introduce detailed explanations in experiments. Also, in this work, we only consider demographic parity, which is one of the most popular definitions of fairness using the independence of the prediction and the sensitive attribute. However, the probable segmentation method can be easily extended to the other fairness metrics. For instance, for equalized odds, which is based on the label-conditional independence of the prediction and the sensitive attribute, we can apply the probable segmentation given the condition $Y$. However, future work remains in this case in that the probable segments for each $Y$ can be different.

## 5 EXPERIMENTS

### 5.1 SYNTHETIC EXPERIMENT

**Dataset**  We experimentally analyze probable demographic parity using a synthetic dataset, following Jiang et al. (2021). The synthetic dataset is generated from the Gaussian distribution $(S, \hat{Y}) \sim \mathcal{N}(\mu, \Sigma)$ with mean $\mu = [0.5, 1.0]$ and variance $\Sigma = \begin{bmatrix} 1.0 & 0.5 \\ 0.5 & 2.0 \end{bmatrix}$. We experiment on the different sizes of the synthetic dataset such as $N = 3000, 5000, 10000, 20000$. Different from a real-world dataset, in which the real distribution is unknown and sampled data samples are only available, the synthetic dataset can be applied to analyze the estimation error using the knowledge of the real distribution.

**Experimental setup**  We verify our design choice of probable segmentation by utilizing equally spaced segmentation which identically segments the sensitive attribute for comparison. We compare the errors of the local prediction averages of probable segmentation and equally spaced segmentation. The local prediction average of the segment is estimated by averaging the predictions of the

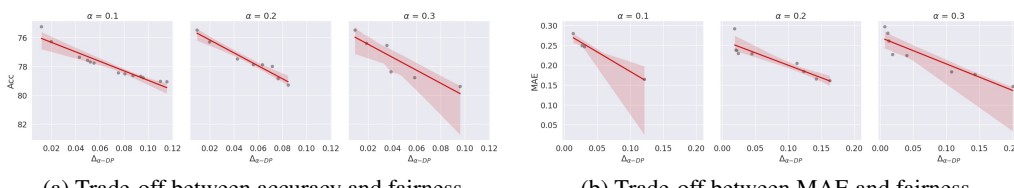

(a) Trade-off between accuracy and fairness          (b) Trade-off between MAE and fairness

Figure 3: Illustrations of the trade-offs between fairness and task performances using the Pareto front. The experiments are repeated 5 times for each fairness regularization hyperparameter $\lambda = \{0.1, 0.5, 1, 5, 50\}$. For measuring the proposed fairness metric $\Delta_{\alpha\text{-}DP}$, $\alpha = 0.1, 0.2, 0.3$ is applied. The experiments (a) for a classification task are conducted using the UCI Adult dataset and the experiments (b) for a regression task are conducted using the Crime dataset.

given samples in the segment, and then the estimation error is computed from the directly calculated real local prediction average.

**Results**    Figure 2 shows the empirical results for the estimation errors of equally spaced segmentation and probable segmentation. For both segmentation methods, a large sample size $N$ results in a stable estimation. At the low probability region from the synthetic dataset such as the left or the right ends of the input, equally spaced segmentation shows the unstable estimations for the local prediction averages. This is natural because the segment in which the insufficient samples are included results in the large variance of the sample mean. However, we observe that probable segmentation, in which the number of samples in each segment is assured, has a robust estimation error across the segments.

## 5.2    REAL-WORLD EXPERIMENTS

**Dataset**    We experiment on the two fairness benchmark datasets, UCI Adult and Crimes, for the real-world binary classification task and the regression task, respectively. UCI Adult dataset (Becker & Kohavi, 1996) consists of 45,555 valid samples annotated with 14 attributes. Among these features, we regard age as a continuous sensitive attribute. UCI Adult dataset aims to classify whether the income exceeds $50K/year, which denotes a binary classification task. Since the proposed fairness metric, probable demographic parity, can be applied to the continuous label, we also investigate its effect on a regression task using the Crime dataset (Redmond, 2009). The goal of the Crime dataset is to predict the number of violent crimes per capita, which denotes a regression task. The Crime dataset consists of 1,994 valid samples annotated with 128 attributes, and we regard the percentage of the population that is African American as a continuous sensitive attribute.

**Experimental setup**    We consider a regularization-based fairness-aware learning method based on Kamishima et al. (2012) as our baseline approach. In addition to the task loss $L$, the fairness metric $\Delta$ is added to the training loss with a regularization scaling hyperparameter $\lambda$; that is, the training loss is equal to $L + \lambda\Delta$. $\lambda$ scales the regularization degree, thereby balancing task performance and fairness. In this work, we directly regularize probable demographic parity using a regularization-based fairness-aware learning method. Two-layer neural networks with 50 hidden units are utilized for both datasets, the UCI Adult dataset and the Crime dataset. As an activation function, SELUs are applied from Klambauer et al. (2017). We optimize the models using Adam optimizer (Kingma & Ba, 2014) with a learning rate of 0.001 and batch size of 256 for 10 epochs. We repeat the experiments five times with different train-test splits.

**Results**    In Figure 3, trade-offs are observed between the proposed fairness metric $\Delta_{\alpha\text{-}DP}$ and task performances. Both the accuracy and MAE are degraded with the enhancement of the fairness metric for every $\alpha = 0.1, 0.2, 0.3$. We further investigate the effect of a regularization hyperparameter $\lambda$. Table 1 shows that there is a trade-off between the accuracy and fairness for each $\alpha$ on the UCI Adult dataset. When $\lambda = 0$, which denotes a training without a fairness regularization, the observed fairness metrics are unsuccessful over every $\alpha$, although it shows the highest accuracy. With the increase of $\lambda$, the model tends to achieve the improved probable demographic parity metric, however, the task performance deteriorates. This remarks that regularization-based fairness-aware

| Adult | $\alpha = 0.1$ | | $\alpha = 0.2$ | | $\alpha = 0.3$ | |
|---|---|---|---|---|---|---|
| | Acc ($\uparrow$) | $\Delta_{\alpha\text{-}DP}$ ($\downarrow$) | Acc ($\uparrow$) | $\Delta_{\alpha\text{-}DP}$ ($\downarrow$) | Acc ($\uparrow$) | $\Delta_{\alpha\text{-}DP}$ ($\downarrow$) |
| $\lambda = 0$ | 78.96 | 0.1311 | 78.96 | 0.1174 | 78.96 | 0.0996 |
| $\lambda = 0.1$ | 78.59 | 0.1051 | 78.41 | 0.0932 | 78.59 | 0.0897 |
| $\lambda = 0.5$ | 78.47 | 0.0915 | 78.11 | 0.0783 | 77.56 | 0.0777 |
| $\lambda = 1$ | 78.55 | 0.0905 | 77.84 | 0.0764 | 77.96 | 0.0703 |
| $\lambda = 5$ | 78.16 | 0.0860 | 78.29 | 0.0831 | 77.46 | 0.0610 |
| $\lambda = 50$ | 77.20 | 0.0580 | 77.36 | 0.0481 | 77.15 | 0.0490 |

Table 1: Binary classification accuracy and the proposed fairness metric, probable demographic parity, on the UCI Adult dataset. $\lambda = 0$ denotes a training without a fairness regularization.

| Crime | $\alpha = 0.1$ | | $\alpha = 0.2$ | | $\alpha = 0.3$ | |
|---|---|---|---|---|---|---|
| | MAE ($\downarrow$) | $\Delta_{\alpha\text{-}DP}$ ($\downarrow$) | MAE ($\downarrow$) | $\Delta_{\alpha\text{-}DP}$ ($\downarrow$) | MAE ($\downarrow$) | $\Delta_{\alpha\text{-}DP}$ ($\downarrow$) |
| $\lambda = 0$ | 14.27 | 0.5139 | 14.27 | 0.3942 | 14.27 | 0.3396 |
| $\lambda = 0.1$ | 17.60 | 0.1825 | 17.54 | 0.1457 | 16.97 | 0.1874 |
| $\lambda = 0.5$ | 25.88 | 0.0336 | 24.05 | 0.0375 | 23.69 | 0.0258 |
| $\lambda = 1$ | 25.22 | 0.0441 | 24.59 | 0.0341 | 25.72 | 0.0242 |
| $\lambda = 5$ | 28.16 | 0.0267 | 28.15 | 0.0355 | 27.47 | 0.0198 |
| $\lambda = 50$ | 31.12 | 0.0381 | 29.35 | 0.0234 | 31.29 | 0.0320 |

Table 2: Mean absolute error (MAE) of a regression task and the proposed fairness metric, probable demographic parity, on the Crime dataset. $\lambda = 0$ denotes a training without a fairness regularization.

learning might be an adequate method for fairness-aware training and also narrates a need for a training method that surpasses the inherent trade-off.

Table 2 presents the experimental results on the Crime dataset. Similar to the UCI Adult dataset, the mean absolute error (MAE) increases with the increasing $\lambda$ only except for the $\lambda = 0.5, 1$ given $\alpha = 0.1$, which indicates the degradation of the regression performance. In contrast, the unfairness tends to be mitigated with increasing $\lambda$ although the fairness performances are not strictly aligned with $\lambda$. This trade-off between task performance and fairness can be directly discovered when $\lambda$ is increased from $0.1$ to $0.5$, in which fairness is enhanced at least relative 74%, and we suffer from the surges of the MAE at least relative 37%. Interestingly, we observe that the largest $\lambda$, i.e., $\lambda = 50$, always shows the worst MAE, but the unfairness problem is not alleviated; rather the fairness metrics suddenly deteriorate where $\alpha = 0.1, 0.3$. Overfitting may account for this result, and the optimal fairness-aware training method remains a future work for our metric.

# 6 CONCLUSION

In this work, we propose a novel definition of fairness, probable demographic parity, for continuous sensitive attributes. The proposed fairness metric measures the maximum disparity of the probable segments' local prediction averages to take advantage of both the mean disparity and the maximum disparity-based fairness metric. Furthermore, we investigate the connection between probable demographic parity and demographic parity, which is the definition of fairness, in various aspects. Theoretical analysis demonstrates the stability of the proposed fairness metric. In addition, we present the empirical version of probable demographic parity and experimentally verify the practical superiority.

**Future works** Our future work is to provide the optimal solution for $\alpha$ upon the experimental and theoretical analysis in this work. Also, since we focus on the features of probable demographic parity in terms of fairness measurement, comparing the previous training methods such as Kamiran et al. (2012); Louppe et al. (2017); Lohia et al. (2019) and designing a novel training method for the proposed metric is promising future work.

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

## A APPENDIX

### A.1 PROOF OF THEOREM 3

*Proof.* Note that $\{seg_{(i)}^\alpha\}_{i=1}^K$ is a partition that satisfies $\bigcup_{i=1}^K seg_{(i)}^\alpha = [0,1)$ and $\forall i,j \in \{1,...,K\}(j \neq i)$, $seg_{(i)}^\alpha \bigcap seg_{(j)}^\alpha = \emptyset$), and we have the fairness metric of $S$ as $\Delta_{\alpha\text{-}DP}(S) = \delta$. Because $seg_{(i)}^\alpha$ is a minimally probable segment by the definition, i.e., $\forall i \in \{1,...,K\}, seg_{(i)}^\alpha \in \{[k, h_\alpha(k))\}$, the following holds:

$$\max_k M_{X,S}([k, h_\alpha(k))) \geq M_{X,S}(seg_{(i)}^\alpha),$$
$$\min_k M_{X,S}([k, h_\alpha(k))) \leq M_{X,S}(seg_{(i)}^\alpha).$$

Thus, $\Delta_{DP}(Q(S))$, which denotes a fairness of $Q(S)$ using demographic parity, satisfies the following:

$$
\begin{aligned}
\delta =& \Delta_{\alpha\text{-}DP}(S) \\
=& \max_k M_{X,S}([k, h_\alpha(k))) - \min_k M_{X,S}([k, h_\alpha(k))) \\
\geq& \max_i M_{X,S}(seg_{(i)}^\alpha) - \min_i M_{X,S}(seg_{(i)}^\alpha) \\
=& \max_i \mathbb{E}_X[f(X)|Q(S) = i] - \min_i \mathbb{E}_X[f(X)|Q(S) = i] \\
=& \Delta_{DP}(Q(S)).
\end{aligned}
$$

Also, because $\forall i \in \{1, ..., K\}, \Pr(Q(S) = i) \geq \alpha$ by the definition of $Q(S)$ which is from a set of minimally probable segment, fairness measured by the violation of demographic parity, $\Delta_{DP}(Q(S))$, and the violation of probable demographic parity, $\Delta_{\alpha\text{-}DP}^d(Q(S))$, with respect to $Q(S)$ are identical by Theorem 1. That is,

$$
\Delta_{DP}(Q(S)) = \Delta_{\alpha\text{-}DP}^d(Q(S)).
$$

Therefore,

$$
\delta \geq \Delta_{DP}(Q(S)) = \Delta_{\alpha\text{-}DP}^d(Q(S)).
$$

$\square$

## A.2 PROOF OF THEOREM 4

*Proof.* Let $L = \max_k |h_\alpha(k) - k|$,

$s_{max} = \arg\max_s \mathbb{E}_X[f(X)|S = s]$, and $s_{min} = \arg\max_s \mathbb{E}_X[f(X)|S = s]$.

Then set any $k_{max}$ satisfying $s_{max} \in [k_{max}, h_\alpha(k_{max}))$.

From the definition of $L$,

$$
[k_{max}, h_\alpha(k_{max})) \subset [s_{max} - L, s_{max} + L) \cap [0, 1).
$$

Using Lipschitz condition, for every $s$ in the segment $[k_{max}, h_\alpha(k_{max}))$, we have

$$
|\mathbb{E}_X[f(X)|S = s] - \mathbb{E}_X[f(X)|S = s_{max}]| \leq \gamma L.
$$

As $\mathbb{E}_X[f(X)|S = s] \geq \mathbb{E}_X[f(X)|S = s_{max}] - \gamma L$, the integration is bounded as

$$
\begin{aligned}
& \frac{\int_{k_{max}}^{h_\alpha(k_{max})} \mathbb{E}_X[f(X)|S = s] \Pr(S = s)ds}{\int_{k_{max}}^{h_\alpha(k_{max})} \Pr(S = s)ds} \\
\geq& \frac{\int_{k_{max}}^{h_\alpha(k_{max})} (\mathbb{E}_X[f(X)|S = s_{max}] - \gamma L) \Pr(S = s)ds}{\int_{k_{max}}^{h_\alpha(k_{max})} \Pr(S = s)ds} \\
=& \mathbb{E}_X[f(X)|S = s_{max}] - \gamma L.
\end{aligned}
$$

Therefore,

$$
\begin{aligned}
\max_{[k, h_\alpha(k))} M_{X,S}([k, h_\alpha(k))) \geq& M_{X,S}([k_{max}, h_\alpha(k_{max}))) \\
\geq& \mathbb{E}_X[f(X)|S = s_{max}] - \gamma L.
\end{aligned}
$$

On the other side, for every $k$,

$$
\begin{aligned}
M_{X,S}([k, h_\alpha(k))) =& \frac{\int_{k_{max}}^{h_\alpha(k_{max})} \mathbb{E}_X[f(X)|S = s] \Pr(S = s)ds}{\int_{k_{max}}^{h_\alpha(k_{max})} \Pr(S = s)ds} \\
\leq& \frac{\int_{k_{max}}^{h_\alpha(k_{max})} (\mathbb{E}_X[f(X)|S = s_{max}]) \Pr(S = s)ds}{\int_{k_{max}}^{h_\alpha(k_{max})} \Pr(S = s)ds} \\
=& \mathbb{E}_X[f(X)|S = s_{max}].
\end{aligned}
$$

Finally,

$$\mathbb{E}_X[f(X)|S = s_{max}] - \gamma L \le \max_{[k,h_\alpha(k))} M_{X,S}([k, h_\alpha(k))) \le \mathbb{E}_X[f(X)|S = s_{max}],$$

leading to

$$|\max_{[k,h_\alpha(k))} M_{X,S}([k, h_\alpha(k))) - \mathbb{E}_X[f(X)|S = s_{max}]| \le \gamma L.$$

Similarly for minimum,

$$|\min_{[k,h_\alpha(k))} M_{X,S}([k, h_\alpha(k))) - \mathbb{E}_X[f(X)|S = s_{min}]| \le \gamma L.$$

From the definition of demographic parity

$$\Delta_{DP} = \max_s \mathbb{E}_X[f(X)|S = s] - \min_s \mathbb{E}_X[f(X)|S = s]$$
$$= \mathbb{E}_X[f(X)|S = s_{max}] - \mathbb{E}_X[f(X)|S = s_{min}],$$

we obtain

$$|\Delta_{DP} - \Delta_{\alpha\text{-}DP}|$$
$$= |(\mathbb{E}_X[f(X)|S = s_{max}] - \mathbb{E}_X[f(X)|S = s_{min})$$
$$- (\max_{[k,h_\alpha(k))} M_{X,S}([k, h_\alpha(k))) - \min_{[k,h_\alpha(k))} M_{X,S}([k, h_\alpha(k))))|$$
$$= |(\mathbb{E}_X[f(X)|S = s_{max}] - \max_{[k,h_\alpha(k))} M_{X,S}([k, h_\alpha(k))))$$
$$- (\mathbb{E}_X[f(X)|S = s_{min}] - \min_{[k,h_\alpha(k))} M_{X,S}([k, h_\alpha(k))))|$$
$$\le |(\mathbb{E}_X[f(X)|S = s_{max}] - \max_{[k,h_\alpha(k))} M_{X,S}([k, h_\alpha(k))))|$$
$$+ |(\mathbb{E}_X[f(X)|S = s_{min} - \min_{[k,h_\alpha(k))} M_{X,S}([k, h_\alpha(k))))|$$
$$\le \gamma L + \gamma L$$
$$= 2\gamma L.$$

$\square$

