# OpenReview forum: "Measuring Fairness Using Probable Segmentation for Continuous Sensitive Attributes"
_ICLR.cc/2024/Conference — Submitted to ICLR 2024_

### Official Review · Reviewer_RKaG · 2023-10-19

**Soundness:** 3 good
**Presentation:** 3 good
**Contribution:** 2 fair
**Rating:** 3
**Confidence:** 4

**Summary:**

This paper proposes probable demographic parity as a generalized measurement of demographic parity on continuous sensitive attributes. The proposed metric measures the maximum prediction parity between segmentations of the sensitive attribute with higher than alpha probability. In this way, it is robust to outliers and noises.

**Strengths:**

1. Fairness with continuous sensitive attributes is relatively under explored. So this is a good topic.

2. The math behind is explained very clearly.

3. Based on the design of the probable demographic parity, I agree that it can be considered as a trade-off between mean prediction disparity and maximum prediction disparity and has the potential to resolve the problems of those two metrics. However, the demonstration of this is lack in the experiments.

**Weaknesses:**

1. The underestimation and overestimation problems in Paragraph 2 were not cleared explained via Figure 1 because neither of mean prediction disparity or maximum prediction disparity were explained before that. Need to adjust the presentation order (present the content in Background first).

2. Very limited related work is presented. E.g. are there literature studying the other metrics such as equalized odds on continuous sensitive attributes? Are mean prediction disparity and maximum prediction disparity the only approaches for continuous sensitive attributes for demographic parity? Part of the discussion in the introduction should be put in the related work section. There are also existing literature using distance covariance to measure independence between the sensitive attributes and the predictions [1, 2]. They also allow continuous sensitive attributes.

3. There is no guidance in what alpha value should be chosen for the proposed metric although the alpha value will greatly impact the metric.

4. The authors have claimed superiority of the proposed metric over mean prediction disparity and maximum prediction disparity. However, there is no experiment demonstrating such superiority. The evaluation has no other baseline.

[1] Liu, Ji, Zenan Li, Yuan Yao, Feng Xu, Xiaoxing Ma, Miao Xu, and Hanghang Tong. "Fair representation learning: An alternative to mutual information." In Proceedings of the 28th ACM SIGKDD Conference on Knowledge Discovery and Data Mining, pp. 1088-1097. 2022.

[2] Guo, Dandan, Chaojie Wang, Baoxiang Wang, and Hongyuan Zha. "Learning Fair Representations via Distance Correlation Minimization." IEEE Transactions on Neural Networks and Learning Systems (2022).

**Questions:**

1. How do you choose the appropriate alpha for the proposed metric?

2. Can you show in the experiments that the proposed metric is better than mean prediction disparity and maximum prediction disparity?

3. What do you think of the distance covariance based metrics from [1] and [2] listed in my Weaknesses when compared to your proposed metric?

4. How can the proposed metric be adapted to multiple sensitive attributes? When the number of sensitive attributes grows, it becomes difficult to find the segmentations.

---

> ### Author Response · Authors · 2023-11-19
> **Author's Response to Reviewer RKaG**
>
> Dear Reviewer RKaG,
>
> We sincerely appreciate your comment.
> In this comment, we aim to answer your questions.
>
> **Q. Comparison to the previous metrics**
>
> A. To the best of our knowledge, there is no paper for continuous sensitive attribute's equalized odds, which means that the continuous sensitive attribute’s fairness has been very limited usage and under-explored. Also, the previous methods for discrete sensitive attributes are not directly applicable. For example, although the fairness of continuous sensitive attributes can be defined and expanded from the references [1, 2], the metric is not computable from the sampled data. This is why we focus on the fairness metric of continuous sensitive attributes.
>
> **Q. Appropriate alpha for the proposed metric**
>
> A. We can choose alpha at a training and inference stage. At first, we experiment the impact of alpha for training. We evaluate the PDP with various alpha for each fixed alpha at the training stage.
>
> |Training \ Inference| alpha=0.1 | alpha=0.2 | alpha=0.3|
> |--|--|--|--|
> | **alpha=0.0** | 13.11 | 11.74 | 9.96|
> | **alpha=0.1** | 9.05 | 8.87 | 7.13 |
> | **alpha=0.2** | 9.15 | 7.64 | 6.59 |
> | **alpha=0.3** | 8.62 | 7.65 | 7.03 |
>
> As a result, choosing the alpha and training using PDP with this alpha leads to a decrease of various PDP metrics for different alpha, which means that the training process is robust for alpha. For example, training the model with alpha=0.1 leads to a decrease in PDP of alpha=0.2 and 0.3, both. That is, for training, choosing alpha is a simple hyperparameter that we can choose using the performance of validation data.
>
> For inference, we need a more exquisite strategy. In a real scenario, a fairness metric is used to evaluating the model’s ability. When the given model satisfies “epsilon > metric” where epsilon is a hyperparameter, the model is said to satisfy epsilon-fairness. So, when the epsilon is fixed, we can find the minimum alpha that satisfies epsilon > PDP, since the PDP tends to get smaller with increasing alpha as shown in Table 1, 2. Finally, we can analyze the results of this process. When we have minimum alpha_m that satisfies epsilon > PDP with alpha_m, our model is fair at least for the segments that have a larger probability than alpha_m. For this reason, alpha is not a fixed hyperparameter, but a meaningful and versatile parameter.
>
> **Q. Comparison of the proposed metric to mean prediction disparity and maximum disparity**
>
> A. At first, Fig 1. (a) and (b) illustrate the disadvantages of mean prediction disparity metric and maximum disparity based metric, respectively. The mean disparity based metric underestimates the low probability segment, which is problematic in that the low probability segment usually evokes inequality. On the contrary, the maximum disparity based metric overestimates the low probability segment, which has unstable estimation. PDP aims to balance the two methods by proposing probable segment, which is guaranteed by the stable variance as in thm 2. Figure 2 also describes the detailed results for this. The upper figures of Figure 2 illustrate the estimation error of every segment. In this figure, we can observe that the estimation error is much worse than the probable segmentation, even though some segments include more than one sample. Thus, we can figure out that the estimation error of maximum disparity based metric would be much worse than PDP.
>
> **Q. Multiple sensitive attributes**
>
> A. We agree that there is a possibility that the computation of probable segmentation would be increased with the growth of the number of sensitive attributes. However, when the samples are finite, we can only consider the segments that are available for the given samples. Thus, we can directly adapt our metric to the multiple sensitive attribute case. Detailed computational complexity and an optimal computation method remain as a future work.
>
> At last, we want to emphasize that fairness for the continuous sensitive attributes have not been sufficiently explored, and our work is meaningful in that we address the obstacles when it comes to extending the fairness metrics for discrete sensitive attributes to the continuous sensitive attributes.

---

> ### Author Response · Authors · 2023-11-23
> **Have your questions and concerns are all addressed?**
>
> Since the end of the discussion period is reaching, we want to check whether all the questions or concerns are addressed. We appreciate for your effort for the review.

---

### Official Review · Reviewer_kbnN · 2023-10-28

**Soundness:** 4 excellent
**Presentation:** 4 excellent
**Contribution:** 3 good
**Rating:** 6
**Confidence:** 5

**Summary:**

The authors address the question of measuring fairness when the sensitive attribute is continuous, e.g. weight or age.

In particular, the authors point out the weakness of the GDP measure introduced by Jiang et al., which is not sensitive to small groups, i.e. when a slice of population s \in [a, b] has small probability measure, its contribution to GDP will be negligible. This is a bad quality for a fairness measure, since small populations are often the source of biased classifiers.

The authors propose to consider a new measure, which better approximates the original DP. Unlike GDP, they sample the segments in a way that makes each of them of significant measure (hyperparameter alpha). They confirm their findings with theoretical analysis, as well as experimentally.

Despite sound motivation and the method proposed, I think there is room for improving this paper. Although the theory presents some insightful results, it does not study analyse the whole problem. Specifically, they show how \tilde{M} fluctuates around M, and then they show how DP-alpha based on M approximates \tilde{M}. However, the two results are not connected. For this particular reason, it is not clear what is a good choice of alpha, there is clearly some trade-off happening between Theorem 2 and Theorem 4, but the reader has to figure it out by themselves. Furthermore, they do not address the question of estimating the segments from the sample, h_alpha are assumed to be known from the true distribution. Finally, your solution is relevant to non-parametric regression with nearest neighbours, and this has to be pointed out, with appropriate references.

I also have a concern regarding Theorem 4. The authors propose that as alpha -> 0, the difference goes to zero. But it is not obvious to the reader, perhaps it is better to keep a potentially weaker result in the main part, and move the original one to appendix. I also think this difference (k - h_a(k)) depends on the distribution and the measure around the point k. I.e. if k at the end of a mode of a distribution, this difference will not go to zero. This nuance has to be highlighted.

**Strengths:**

Good motivation and feasible proposed solution.

**Weaknesses:**

Incomplete theory

No validation procedure for choice of alpha proposed

**Questions:**

How do you choose alpha?

Do you need to simultaneously control the difference between M and \tilde{M} for more than one point k?

What are the conditions that the RHS in Theorem 4 goes to 0 as alpha -> 0 and at what rate does it converge?

---

> ### Author Response · Authors · 2023-11-20
> **Author's Response to the Reviewer kbnN**
>
> Dear Reviewer kbnN,
>
> We are so grateful for your positive comments and thoughtful feedback on our work. We have the official response for the review.
>
> **Q. Appropriate alpha for the proposed metric**
>
> A. We can choose alpha at a training and inference stage. At first, we experiment the impact of alpha for training. We evaluate the PDP with various alpha for each fixed alpha at the training stage.
>
> |Training \ Inference| alpha=0.1 | alpha=0.2 | alpha=0.3|
> |--|--|--|--|
> | **alpha=0.0** | 13.11 | 11.74 | 9.96|
> | **alpha=0.1** | 9.05 | 8.87 | 7.13 |
> | **alpha=0.2** | 9.15 | 7.64 | 6.59 |
> | **alpha=0.3** | 8.62 | 7.65 | 7.03 |
>
> As a result, choosing the alpha and training using PDP with this alpha leads to a decrease of various PDP metrics for different alpha, which means that the training process is robust for alpha. For example, training the model with alpha=0.1 leads to a decrease in PDP of alpha=0.2 and 0.3, both. That is, for training, choosing alpha is a simple hyperparameter that we can choose using the performance of validation data.
>
> For inference, we need a more exquisite strategy. In a real scenario, a fairness metric is used to evaluating the model’s ability. When the given model satisfies “epsilon > metric” where epsilon is a hyperparameter, the model is said to satisfy epsilon-fairness. So, when the epsilon is fixed, we can find the minimum alpha that satisfies epsilon > PDP, since the PDP tends to get smaller with increasing alpha as shown in Table 1, 2. Finally, we can analyze the results of this process. When we have minimum alpha_m that satisfies epsilon > PDP with alpha_m, our model is fair at least for the segments that have a larger probability than alpha_m. For this reason, alpha is not a fixed hyperparameter, but a meaningful and versatile parameter.
>
> However, it is still true that the estimation is stable under small alpha. One heuristic solution for this is to control the epsilon with the alpha. As an example, when we set epsilon = 0.1 for alpha = 0.1 as a fairness criterion for our task, using the larger epsilon such as 0.2 for larger alpha as 0.2 would be beneficial concerning the estimation stability.
>
> **Q. Detailed discussion on Thm 4**
>
> A1. We additionally explore the specific condition that the RHS of Thm 4 goes to 0 as alpha -> 0. Suppose that the probability density function has a lower bound L_p. Then, |h_alpha(k) - k| <= alpha/L_p holds because there is a lower bound for the probability density function. Finally, we can find out that if alpha -> 0, the RHS goes to 0. In short, if there exists a lower bound L_p of a probability density function, the RHS of Thm 4 goes to 0 as alpha -> 0.
>
> A2. Here, we also want to respond to your other concern related to the Thm 4. As you noted, it is correct that the bound of Thm 4 is derived from the true distribution. However, we think it is still meaningful since our contribution is that proving the difference between DP and PDP is bounded and reveals the detailed bound from the true distribution, even if it is impossible to estimate the bound from the sampled data directly.
>
> **Q. Do you need to simultaneously control the difference between M and \tilde{M} for more than one point k?**
>
> A. Since PDP is based on the maximum disparity of M, we consider various k on the algorithm to find out the maximum disparity estimation.
>
> In addition, we are thankful for pointing out the related references. We consider 1) Tibshirani, Ryan. "Nonparametric Regression: Nearest Neighbors and Kernels., 2) Hastie, Trevor, et al. The elements of statistical learning: data mining, inference, and prediction. Vol. 2. New York: springer, 2009., and 3) Wasserman, Larry. All of nonparametric statistics. Springer Science & Business Media, 2006. as the reference and reflect these on our final version.

---

> > ### Comment · Reviewer_kbnN · 2023-11-22
> >
> > Thank you for your prompt response, and I appreciate you showing additional experimental results.
> >
> > Could you please clarify what is the training and inference mean in the table? What is being trained in this case?
> >
> > >  So, when the epsilon is fixed, we can find the minimum alpha that satisfies epsilon > PDP, since the PDP tends to get smaller with increasing alpha as shown in Table 1, 2.
> >
> > I do not think it is appropriate to fit $\alpha$ to satisfy the PDP constraints. Alpha should be chosen based on the smoothness properties of $m(s) = E [ f(S) | S= s] $, in other words, it should depend on the smoothness of ground truth. Perhaps some existing non-parametric adaptation techniques can help.

---

> ### Author Response · Authors · 2023-11-23
> **Official response to the comment**
>
> We want to express our gratitude to your effort to improve the paper. There are the answers for your questions below.
>
> **Additional information for alpha in the table**
> There are two types of alpha that we can choose, because we trained the model using the loss function L + r*PDP_a1 and tested using PDP_a2. In the original manuscript, we used identical a at the training and inference stage, i.e., a1=a2. So. we experimented when a1!=a2. Even though the a1=0.2, PDP is decreased at a2=0.1, 0.2, and 0.3, which means that the choice of a1 at training stage is not sensitive.
>
> **Additional opinion for the choice of a**
> There are two perspective for selection of a.
> - The first perspective is related to the robustness of PDP and difference to the DP. Since low a leads to high variance and small max|h_a(k) - k|, there is a trade-off. In here, we can leverage a heuristic for a when the true distribution is unknown. Firstly, we can estimate the E_a:=max|h_a(k) - k| with growing and linear spaced a such as a=(0.1, 0.2, 0.3, … max_a) and a=(0.05, 0.1, 0.15 .., max_a). From this, we can compute E_a[i] - E_a[i+1]. As you suggested, we would choose a where the difference is drastic, because it means that max|h_a(k) - k| is drastically improved and so the bound of Thm 4 is effected.
> - The second perspective is the meaning of PDP itself, as we argued at the previous response. PDP_a < e denotes that even though the model is not fair (Y_h is not independent to S), the model is fair at least the segments with the probability a (and we can derive Y_h is independent to quantized S, as Thm 3). In this case, the variance would be internal risk of the data.

---

### Official Review · Reviewer_xUa4 · 2023-11-01

**Soundness:** 3 good
**Presentation:** 3 good
**Contribution:** 2 fair
**Rating:** 5
**Confidence:** 4

**Summary:**

This work proposes a new measure of fairness called probable demographic parity with a focus on continuous sensitive attributes. The proposed measure is based on a maximum prediction disparity of probable segments. They consider probable segments of the continuous sensitive attribute that have a higher probability than the minimum probability condition.

DP for discrete attributes is defined as Pr(Y=1|S=s)-Pr(Y=1|S=s').
A possible relaxed metric (that would also apply to the continuous case) is defined as: max_s Pr(Y=1|S=s)- min_s'Pr(Y=1|S=s').
Other related metrics are Generalized DP. The measure proposed by the authors looks at the maximum/minimum of the expected value of Y over an interval such that the interval has a measure alpha>0 (scaled down by the probability measure in that interval).

They compare the local prediction average of these segments to identify the maximum prediction disparity. They analyze the various theoretical properties including stability and independence and experimentally demonstrate the benefits of the proposed metric.

**Strengths:**

This paper introduces an interesting extension of demographic parity (DP) for continuous sensitive attributes.
The measure proposed by the authors looks at the maximum/minimum of the expected value of Y over an interval such that the interval has a measure alpha>0 (scaled down by the probability measure in that interval).
For discrete attributes, this definition reduces to DP as shown in Thm 1.

Then, they also employ empirical estimation techniques to compute the proposed measure which has benefits in terms of computational complexity. They also include additional theoretical results on how far the DP will be to Probable DP for Quantized sensitive attributes and also under Lipschitz assumptions.

**Weaknesses:**

Thm 1 statement alpha>0?

The experiment results only seem to demonstrate that the algorithm works and the measure is computable. However, not much insight is provided on how this way of computation is beneficial.

Is there a distribution with ground truth DP and PDP known so that the performance of the estimation can be compared to it?

There is very little detail on Figure 2.

How is this definition better than the Generalized DP? Could the authors elaborate on this?

There is no discussion on the limitations of this strategy/definition.

While the idea is interesting, I feel that given other related works, this work is of limited novelty in the context of related works.

**Questions:**

Included in Weakness

---

> ### Author Response · Authors · 2023-11-19
> **Author's Response to the Reviewer xUa4**
>
> Dear Reviewer xUa4,
>
> We thank you for your comment and interest in the paper. We want to answer the questions and clarify your concerns in this comment.
>
> For discrete sensitive attributes, the maximum disparity-based metric is a popular choice. However, directly applying this metric for continuous sensitive attributes is not possible or suffers from estimation instability. In this situation, our work is meaningful in that we propose a computable novel metric while handling previous work’s issues. We also show that PDP is also closely related to DP for discrete case and continuous case both in Theorem 1, 3, 4.
>
> Specifically, we address the disadvantages of GDP in terms of 1) **estimation error** and 2) **underestimation problem**.
>
> Experimentally, we explore the estimation stability in section 5.1, and the results are shown in Fig 2. For the synthetic Gaussian distribution case, we compare the estimation error of equally spaced segments, which is applied to compute GDP to the estimation error of probable segment, which is applied to our metric PDP. Equally spaced segmentation denotes the estimation method that segments the sensitive attribute by an equal length. GDP based on equally spaced segmentation can suffer from high estimation error for the low probability regions such as the extreme parts of the Gaussian distribution, which is clearly shown in Figure 2. We can see that the estimation error of the equally spaced segmentation method is relatively large at the sensitive attribute’s low probability region. Contrary to this, PDP based on probable segmentation shows stable errors across the segments. In theory, this is easily expected because by theorem 2, the variance of each segment in PDP is bounded with \sigma^2/N\alpha, but in GDP, the bound does not exist.
>
> The second point of view is the underestimation problem. GDP might ignore the low probability segment since GDP is based on a mean disparity-based metric. This is a vulnerable characteristic as a fairness metric because the inequality tends to originate from low probability regions. Fig 1. (a) illustrates this problem where the inequality of a low probability segment is underrepresented by the mean dispairty-based metric such as GDP.
>
> We also explore an additional feature of PDP. When training the model with PDP using a fixed alpha, we observe that PDP metrics using other alpha are also decreased. For example, when we train PDP with alpha=0.1, PDP with alpha=0.2 is decreased from 11.74 to 9.39 and PDP with 0.3 is also decreased from 9.96 to 8.04. This also holds for training using PDP with alpha=0.2, 0.3. That is, it shows the robustness of selecting alpha at the training stage.
>
> Although the training the model with PDP has this advantage, one of the main limitations of PDP is that training process is not guaranteed. Even though we experiment PDP using regularization-based fairness learning, the metric does not give us a fair training method for optimal solution. This should be further explored and elaborated.
>
> Additionally, we add the condition that alpha>0 for Thm 1.
>
> At last, we want to emphasize that fairness for the continuous sensitive attributes have not been sufficiently explored, and our work is meaningful in that we address the obstacles when it comes to extending the fairness metrics for discrete sensitive attributes to the continuous sensitive attributes.

---

> ### Author Response · Authors · 2023-11-23
> **Have your questions and concerns are all addressed?**
>
> Since the end of the discussion period is reaching, we want to check whether all the questions or concerns are addressed. We appreciate for your effort for the review.

---

### Meta-Review · Area_Chair_6Nei · 2023-12-09

**Metareview:**

The authors explore the challenge of measuring fairness in scenarios where the sensitive attribute is continuous, such as weight or age. They specifically critique the GDP measure introduced by Jiang et al., highlighting its insensitivity to small groups. When a population slice, denoted as $$s \in [a,b]$$, has a small probability measure, its contribution to GDP becomes negligible. This deficiency is problematic for a fairness measure, as small populations often contribute to biased classifiers. To address this issue, the authors propose a new measure that provides a more accurate approximation of the original DP. In contrast to GDP, they adopt a sampling approach that ensures each segment has a significant measure, controlled by the hyperparameter alpha. The authors support their proposal through theoretical analysis and experimental validation.

While the reviewers acknowledge that the problem is well motivated, and the proposal looks interesting, they raised concerns on the benefits of the proposed solution. The authors are encouraged to perform more substantial experiments to justify the advantage of adopting their solution over several baselines. One reviewer points out that there are some gaps in the presented theoretical results, making it harder for the readers to have a full picture of the achieved theoretical guarantees.

**Justification For Why Not Higher Score:**

This paper has potential but the current state of it would require another revision. The reviewers have identified several things the authors want to add to the next version, including more baseline comparisons, comparisons to the gold standards, better consolidating and positioning the theoretical results.

**Justification For Why Not Lower Score:**

N/A

---

### Decision · Program_Chairs · 2024-01-16

Reject